# The Prediction of Influenza-like Illness and Respiratory Disease Using LSTM and ARIMA

**DOI:** 10.3390/ijerph19031858

**Published:** 2022-02-07

**Authors:** Yu-Tse Tsan, Der-Yuan Chen, Po-Yu Liu, Endah Kristiani, Kieu Lan Phuong Nguyen, Chao-Tung Yang

**Affiliations:** 1Department of Emergency Medicine, Taichung Veterans General Hospital, Taichung 407204, Taiwan; janyuhjer@gmail.com; 2School of Medicine, Chung Shan Medical University, Taichung 40201, Taiwan; 3Division of Occupational Medicine, Department of Emergency Medicine, Taichung Veterans General Hospital, Taichung 407204, Taiwan; 4College of Medicine, China Medical University, Taichung 406040, Taiwan; dychen1957@gmail.com; 5Rheumatology and Immunology Center, China Medical University Hospital, Taichung 404332, Taiwan; 6Division of Infection, Department of Internal Medicine, Taichung Veterans General Hospital, Taichung 40704, Taiwan; pyliu@vghtc.gov.tw; 7Department of Computer Science, Tunghai University, Taichung 407224, Taiwan; endahkristi@thu.edu.tw; 8Department of Informatics, Krida Wacana Christian University, Jakarta 11470, Indonesia; 9Faculty of Environmental and Food Engineering, Nguyen Tat Thanh University, Ho Chi Minh City 70000, Vietnam; nklphuong@ntt.edu.vn; 10Research Center for Smart Sustainable Circular Economy, Tunghai University, No. 1727, Sec. 4, Taiwan Boulevard, Taichung City 407224, Taiwan

**Keywords:** air pollution, PM2.5, ILI, influenza-like illness, respiratory disease, LSTM, ARIMA

## Abstract

This paper proposed the forecasting model of Influenza-like Illness (ILI) and respiratory disease. The dataset was extracted from the Taiwan Environmental Protection Administration (EPA) for air pollutants data and the Centers for Disease Control (CDC) for disease cases from 2009 to 2018. First, this paper applied the ARIMA method, which trained based on the weekly number of disease cases in time series. Second, we implemented the Long short-term memory (LSTM) method, which trained based on the correlation between the weekly number of diseases and air pollutants. The models were also trained and evaluated based on five and ten years of historical data. Autoregressive integrated moving average (ARIMA) has an excellent model in the five-year dataset of ILI at 2564.9 compared to ten years at 8173.6 of RMSE value. This accuracy is similar to the Respiratory dataset, which gets 15,656.7 in the five-year dataset and 22,680.4 of RMSE value in the ten-year dataset. On the contrary, LSTM has better accuracy in the ten-year dataset than the five-year dataset. For example, on average of RMSE in the ILI dataset, LSTM has 720.2 RMSE value in five years and 517.0 in ten years dataset. Also, in the Respiratory disease dataset, LSTM gets 4768.6 of five years of data and 3254.3 of the ten-year dataset. These experiments revealed that the LSTM model generally outperforms ARIMA by three to seven times higher model performance.

## 1. Introduction

The primary factor of global death and disease leads to outdoor air pollutants. The increasing number of hospital and emergency room visitations indicates when environmental exposures happened indicates that there is a correlation between them [1]. World Health Organization (WHO) addressed that around 4.2 million premature deaths cases are correlated with air pollution [2,3]. The cases account for 29% from lung cancer, 17% from an acute lower respiratory infection, 24% from stroke, 25% from ischaemic heart disease, and 43% from chronic obstructive pulmonary disease. Therefore, the study of modeling the correlation of air pollutants and diseases such as Influenza-like Illness (ILI) and respiratory illness is notable [4].

This paper proposed the prediction model of air pollutants, ILI, and respiratory disease using Long short-term memory (LSTM) and Autoregressive integrated moving average (ARIMA). This paper aims to predict the amount case of ILI and respiratory disease based on air pollutants parameters. In this case, we compare the different datasets of five years and ten years terms. Also, we applied feature selection methods for the training model of LSTM [5]. The specific objectives are listed as follows:We empirically study the training model of ARIMA and LSTM with the input of air pollutants, and the output is the number of diseases predicted.We Evaluate and compare the training model of ARIMA and LSTM for air pollutant and diseases cases.

This paper’s contribution is to empirically study the comparison models of ILI and Respiratory disease in two different time-spans of 5 and 10 years datasets. Moreover, we compare the trained models using three feature selections from matrix correlation, extra trees classifier, and chemical-based methods. In addition, we also trained all parameters of the air quality dataset. Therefore, we have 20 models of ILI disease and 20 models of Respiratory disease for evaluating the model’s performance. Furthermore, other research might refer to our results for further development. However, this paper has the limitation as follows.
We trained ARIMA, which is the statistical-based and univariate input.We trained LSTM, the neural networks-based, and multivariate input.We implemented three feature selections on LSTM models.We just compared the result of ARIMA and LSTM models.

## 2. Background Reviews and Related Works

### 2.1. Root Mean Squared Error (RMSE)

The RMSE is the standard deviation of residuals (prediction errors). The residuals are a metric about how far these data points are from the regression line, and the RMSE reflects how wide the spread out of these residuals are. In other words, it indicates how narrowly the information is concentrated on the best-fit line. Root mean square error is commonly used to assess experimental findings in climatology, forecasting, and regression analysis. The following is a description of the equation:(1)$$RMSE=\left(1/X\right)\sum_{X}^{\left(i=1\right)}{\left(z_{i}-\cap z_{i}\right)}^{2}$$
where *X* is sample size, $z_{i}$ is the actual expected output, and $\cap z_{i}$ is the model’s prediction.

### 2.2. Feature Selections

Before conducting to analyze time-series data, the input data should be prepared to achieve the best performance. Data pre-processing might include data transformation, handling with missing data, and feature selection. In addition, feature selection is a critical step to identify significant features for noise filtration purpose leading to reduce the dimensionality of the data. This procedure can also improve the accuracy of the model Panda et al. [6]. Several studies propose some approaches to select essential factors to improve time-series forecasting. Zhai and Chen (2018) [7] apply four different feature extraction involving simplification, polynomial, transformation, and combination. As a result of the four methods, NO_2_ and CO contribute the most in forecasting PM_2.5_ concentrations. Sethi and Mittal (2019) [8] denote measuring the relationship between air pollutants and PM_2.5_ play a vital role in predict accurately the concentration of PM_2.5_. The dependencies among features can also decide the relevance of the feature in the dataset, determined using linear regression. The association between PM_2.5_ and PM_10_, and NO_2_, SO_2_ are also measured by Pearson correlation coefficients in a study at 31 Chinese cities (Xie et al., 2015) [9].

### 2.3. Air Pollutants and Public Health Correlations

There is a strong association between deteriorating air quality and public health problems [10,11]. According to the WHO, the exposure to indoor air pollution causes approximately 4.2 million deaths worldwide every year, while the number of deaths relating to ambient air pollution is 3.8 million (World Health Organization, 2016) [12]. The effect of air pollutants has been demonstrated in previous studies on respiratory, nervous, and cardiovascular systems [13]. For example, nitrogen oxides (NOx), sulfur dioxide (SO2), ozone (O3), fine-particular matter (PM2.5) are influencing factors that lead to air pollution-related diseases such as asthma, ischemic heart disease, and neurodevelopmental disorders (Genc et al., 2012 [14]; Kampa and Castanas, 2008 [15]; Özkaynak et al., 2013) [16]. A combination of SO2, NO2, and some heavy metals will result in nose and throat irritation and could become more serious for asthmatic people because of bronchoconstriction and dyspnea symptoms. Gianfrancesco et al. [17] proposed the big data analysis such as the evaluation of the influence of air pollution on ILI requires an adaptation of machine leaning approach. However, the machine learning approach used in their study may also be subject to bias.

### 2.4. ARIMA and LSTM

ARIMA and LSTM are the conventional statistical and deep learning models, respectively, widely applied for analyzing and forecasting time series data in various sectors. Eymen and Köylü (2019) [18] use ARIMA tool to predict relative humidity, which was slipped into pre-dam and post-dam. Pre-dam relative humidity fitted the ARIMA model, while post-dam relative humidity is compatible with data predicted by ARIMA. In addition, Chakraborty et al. (2019) [19] propose hybrid models that integrate ARIMA and some models of machine learning such as artificial neural networks (ANN), support vector machines (SVM), LSTM, and neural network autoregressive (NNAR) applying to dengue time-series data. In the business and finance field, Siami-Namini and Namin (2018) [20] investigate how performance of ARIMA and LSTM in stock data of some companies between Jan 1985 and Aug 2018. Consequently, by comparing the RMSE metric, LSTM is confirmed to outperform the statistical-based algorithm, ARIMA. Due to the difficulty of obtaining parameters hydrogeological characteristics, Zhang et al. (2018) [21] proposes an approach that combines a fully connected layer associated with the dropout method on the top and LSTM layer to predict water table depth in the agricultural area. The result shows a higher R2 score comparing to a model with double LSTM layers [22]. Similarly, Kristiani et al. (2020) [23] claim a high correlation between PM2.5 and NO2, SO2 and prove an improved accuracy of forecasting PM2.5 by sequence to sequence LSTM model.

### 2.5. Related Works

The air quality and disease cases dataset are belonging to spatiotemporal time-series data, which is a series of data points in time order and specific geographical (or spatial) points [24,25]. Spatial variation in disease cases has also demonstrated dependence patterns and noise levels in the data. Spatiotemporal analyzes have additional advantages as they simultaneously learn pattern persistence over time and investigate anomalies behavior [26]. The ability to interact with space and time might also detect data clustering, which leads to discover the emerging of environmental hazards [9,27]. The analytic results by using LSTM and ARIMA may aid in making early detection of IL1 and optimal preventive strategies for air pollution [28,29]. A comparative study between a model applies an ensemble approach with and without feature selection to predict the different classes of skin disease by Verma et al. (2019) [30]. An ensemble method combines six classification algorithms in order including passive-aggressive classifier, linear discriminant analysis, radius neighbor classifier, Bernoulli naive Bayesian, Gaussian naive Bayesian, and Extra Trees Classifier. The findings show efficient use in the detection of erythemato-squamous diseases with improvement in speed and accuracy.

## 3. Material and Methods

In this section, the system design of the system is presented in three subsections of the system architecture, the experimental workflows, and the parameter used in this paper.

### 3.1. System Environment

The specific environment is required to provide machine learning experiments. The dataset of air pollutants, ILI, and respiratory disease are trained using ARIMA and LSTM. TensorFlow’s deep learning library was used to train air quality and disease history integration data. The library used Numpy and Pandas to process data manipulation and preprocessing. Matplotlib was used to visualize real and predictive information. Scikit-learn applied to evaluate the model’s accuracy. Figure 1 describes the architecture of the environmental scheme used in the project.

### 3.2. System Workflow

Figure 2 describes the experimental procedure. First, the data were collected from the Taiwan Environmental Protection Administration (EPA) and The Taiwan Centers for Disease Control (CDC). Then, we did the data preprocessing by filling the missing data with 0 value and removing the outliers. The dataset was then divided into two categories of five years and ten years. For ARIMA model training, the data used weekly disease cases. For LSTM model training, we correlated the dataset with air pollutants. Therefore, we applied feature selection. There are three methods of feature selection that is Matrix Correlation (MC), Extra Trees Classifier (ETC), and Chemical (CHE) base. After the feature selection process, we train the dataset in five models of ARIMA, LSTM MC, LSTM ETC, LSTM CHE, and LSTM all Parameters.

### 3.3. Experimental Procedure

The experimental procedures are described in this section from data preprocessing, Feature Selection, ARIMA, and LSTM parameters.

#### Data Preprocessing

Two datasets are extracted from EPA [31] and CDC [32] from the year 2009 to 2018. There are two kinds of dataset, air quality/pollutants, and the number of ILI [33] and respiratory cases [34] in weekly from insurance resource. All subjects gave their informed consent for inclusion before they participated in the study. From these two datasets, we integrated the data and did data preprocessing, as follows:Convert the air quality dataset from hourly into daily by averaging all the parameters.Combine the air quality dataset daily with ILI and Respiratory dataset then convert onto weekly data for training process.Replace the missing data and the outliers by zero value.

In this case, we do not use all AQI parameters as three parameters of several stations are not available. Therefore, we removed three parameters, THC, CH4, and NMHC, to make the training process equally. All 15 parameters of air pollutant are described in Section 3.4.4. The detailed extracted dataset is 261 records for 5 years and 520 records for 10 years for the training process.

### 3.4. Feature Selection

Before the training process of LSTM, the feature selection is applied for dropping the number of input parameters when developing a predictive model. In this case, we implemented three categories of feature selection and 15 air pollutant parameters as follows:

#### 3.4.1. Matrix Correlation

A matrix correlation is a table representing the association coefficients for several variables. Any random variable in the table is strongly connected to each of the other values. It allows us to see which couples have the uppermost correlation [35]. Table 1 shows the 5 parameters feature selection of matrix correlation of ILI and Respiratory diseases.

#### 3.4.2. Extra Trees Classifier

Extremely Randomized Trees Classifier is a method of ensemble learning strategy which consolidates the results of multiple de-correlated decision trees obtained in a forest to generate its classification outcome. In concept, it is very similar to a Random Forest Classifier and only differs from it in the way decision trees are built in the forest. Table 2 shows the 5 parameters feature selection of Extra Trees Classifier of ILI and Respiratory diseases.

#### 3.4.3. Feature Selection of Chemical Base

In general, PM2.5 can be separated into primary and secondary aerosols inside the atmosphere. Primary aerosols are generated directly by natural and anthropogenic contaminants, while the chemical reactions are occurring in the atmosphere release secondary aerosols. Secondary aerosols influence existing primary aerosols and relevant gaseous precursors. The development of secondary PM2.5 is closely linked to precursor gases NO2 and SO2. This association is illustrated in Lee et al. [36]’s 2005–2015 Taiwan Air Quality Studies. The U.S. Environmental Protection Agency also suggested using SO2, NO2/NOx, to assess the formation of secondary fine particulate matter (PM2.5) for modeling air dispersion. Table 3 listed the chemical correlation of feature selection for ILI and Respiratory disease.

#### 3.4.4. All Parameters of Air Pollutant Dataset

The dataset extracted from the Taiwan Environmental Protection Administration (EPA) for the years 2009–2018 (10 years data). It consists of five stations, Xitun, Chungming, Fengyuan, Dali, and Shalu in Taichung City Taiwan. The features include 15 parameters that consists of 8 chemical and 7 meteorological pollutant factors [23], as described in Table 4.

### 3.5. Training Parameters

There are two kinds of model training methods, that is ARIMA and LSTM. In this case, we intend to compare the model accuracy of these two methods in a particular environment.

#### 3.5.1. ARIMA Model Training

An ARIMA model is used to determine if ILI and respiratory disease numbers could be predicted in advance. The case of the weekly illness is included in the dataset to predict the trend in the time series. In setting up the ARIMA model, the first 80% of data collected was utilized as training data, while the rest was used as validation data for creating the ARIMA model. The plots of autocorrelation and partial autocorrelation are then made. A *p*-value of less than 0.05 indicates that the null hypothesis of non-stationarity is rejected when a Dickey-Fuller test is performed. The pyramid library’s auto-arima is then used to execute an ARIMA model. It was used to find the ARIMA model’s best (p,d,q). ARIMA Training Procedures are described in Algorithm 1.
**Algorithm 1** ARIMA Training Methods**Require:** ILI, Respiratory, and AQI Dataset integration 
**Ensure:** Sum up the number of disease cases per week and order them
   1:tseriesdt=pd.Series(tseriesdt)   2:size=int(len(tseriesdt)∗0.8)   3:train,val=tseriesdt[0:size],tseriesdt[size:len(tseriesdt)]←Training and Validation data partition   4:p−value<0.05←Execute Dickey-Fuller test to create *p*-value   5:auto_arima(train, start_p=0, start_q=0, max_p=10, max_q=10, start_P=0, start_Q=0, max_P=10, max_Q=10, m=52, seasonal=True, trace=True, d=1, D=1, error_action=’warn’, suppress_warnings=True, random_state = 20, n_fits=30)←
Running the ARIMA model   6:AIC←Get Akaike’s Information Criterion (AIC)   7:RMSE←Calculate the RMSE


Figure 3 describes the SARIMAX(1, 1, 1) × (1, 1, 1, 52) configuration was identified as the most optimal for modeling the time series based on the lowest AIC.

#### 3.5.2. LSTM Model Training

LSTMs are sequential neural networks that assume dependence between the observations in a given series. Such as, they have been widely used for forecasting purposes of the time series. In configuring the LSTM model, the first 100 observation data from the time series dataset was called. After that, a dataset matrix is produced. The data is then adjusted using MinMaxScaler so that the neural network can appropriately perceive it. With the previous option set to 5, the data is divided into training and test sets. When the “previous” parameter is set to this, the values t−1, t−2, t−3, t−4, and t−5 (all under X train) are used to forecast the value at time *t* (Y train for the training data). Following that, 150 epochs are completed. LSTM Training Procedures are described in Algorithm 2.

Figure 4 describes the layer’s structure of LSTM model used in this paper.

The training models are divided by four conditions of feature selection, as follows:Matrix CorrelationExtra Trees ClassifierChemical reactions happening in the atmosphereAll parameters (15 parameters of air pollutant)
**Algorithm 2** LSTM Training Methods**Require:** ILI, Respiratory, and AQI Dataset integration 
**Ensure:** Sum up the number of disease cases per week and order them
  1:df=df[:100]←Call the first 100 observation data  2:MinMaxScaler←Normalize dataset with MinMaxScaler  3:trainsize=int(len(df)∗0.8)  4:valsize=len(df)−trainsize  5:train,val=df[0:trainsize,:],df[trainsize:len(df),:]←Training and validation data partition  6:previous=5←Number of previous parameter  7:X_train,Y_train=create_dataset(train,previous)  8:X_val,Y_val=create_dataset(val,previous)  9:model=tf.keras.Sequential()←Create LSTM model10:model.add(LSTM(4,input_shape=(1,previous)))←Add LSTM layers with 4 hidden layers11:model.add(Dense(1))←Add Dense Layer12:model.compile( loss=’mean_squared_error’, optimizer=’adam’)←Compile the model using Adam optimizer13:model.fit( X_train, Y_train, epochs=150, batch_size=1, verbose=2)←Train the model in 150 epochs14:RMSE←Calculate the RMSE


## 4. Results

In this section, the experimental results are presented which is comparing the forecasting result between ARIMA and LSTM.

### 4.1. ARIMA Prediction Results

With 80% of the time series dataset used as the training data to build the ARIMA model, the remaining 20% is used to test the predictions of the model. Table 5 describes the comparison of RMSE value of four ARIMA models. The table shows that ARIMA models using five years data have a lower RMSE value at 2174.8 and 5581.6, compared to 10 years of data usage at 8173.6 and 15,093.1.

The following tables describe the actual data compared to validation and prediction. The blue line represented the real data used for the training. The orange line is the validation for the forecast, and the green line shows the prediction of the validation data. It can be seen from the graph’s trends, The trend’s line in Figure 5a has the closed prediction compared to Figure 5b–d. It is because of the model in Figure 5a has smallest RMSE value.

### 4.2. LSTM Prediction Results

In the LSTM training models, the feature selections are applied with five times repetitions. For example, Table 6 presented the ILI RMSE in 5 and 10 years data. The data in 10 years has more low error than five years data. The mean of all data, MC, ETC, and CHE for ten years dataset have the RMSE values at 506.2, 473.5, 573.6, and 515.1, respectively. While in 5 years dataset, the RMSE increased about 30% at 730.6, 707.9, 728.7, and 713.6, respectively.

Table 7 described the Respiratory RMSE in 5 and 10 years data. Like in ILI case, The data in 10 years has more low error than five years data. The mean of all data, MC, ETC, and CHE for ten years dataset have the RMSE values at 3258, 3064.7, 3186.7, and 3507.7, respectively. While in 5 years dataset, the RMSE increased about 25% at 4752.1, 4750.3, 4798.1, and 4773.7, respectively.

From the five times repetition, we can draw the boxplot to present the distribution of RMSE values. A boxplot is a standardized way of displaying the distribution of data based on a five number summary (“minimum”, first quartile (Q1), median, third quartile (Q3), and “maximum”). It can also tell about the outliers and what their values are. Figure 6 describes the distribution of the data in boxplot.

### 4.3. Model Comparison

The entire models comparison are presented in the graph Figure 7a,b, as follows.

It can be seen from both graphs, Univariate ARIMA have significant error rate than multivariate LSTM models. ARIMA RMSE values have 4 to 10 times more high compared to LSTM.

### 4.4. Sampling and Result Analysis

In this section, we demonstrate the several prediction results of ARIMA and LSTM models. For ARIMA, we take the example of two different models. First, Arima ILI model prediction of five years dataset with RMSE value at 2564.9 in Figure 8a. Second, ARIMA Respiratory model prediction of ten years dataset with RMSE value at 22680.4 in Figure 8b.

For the LSTM model, we pick up two different models. First, LSTM ILI model prediction of five years dataset with RMSE value at 473.5 in Figure 9b. Second, the LSTM Respiratory model prediction of five years dataset with RMSE value at 4798.1 in Figure 9b. The training and evaluation of each models are described in Figure 9a and Figure 10a.

## 5. Discussion

From the training model experiments, it can be seen that ARIMA has an excellent model in the five-year dataset of ILI at 2564.9 compared to ten years at 8173.6 of RMSE value. This accuracy is similar to the Respiratory dataset, which gets 15,656.7 in the five-year dataset and 22,680.4 of RMSE value in the ten-year dataset. On the contrary, LSTM has better accuracy in the ten-year dataset compared to the five-year dataset. On average RMSE in the ILI dataset, LSTM has 720.2 RMSE value in five years and 517.0 in ten years dataset. Also, in the Respiratory disease dataset, LSTM gets 4768.6 of five years of data and 3254.3 of the ten-year dataset. In terms of the LSTM various parameters to be trained using feature selection, there are no significant differences among the RMSE values. These analyses revealed that the LSTM model typically outperforms ARIMA by three to seven times higher accuracy.

From the ILI boxplot of 5-year and 10-year period data, it can be seen that the MC model in the two periods has a minor interquartile range showing the most stable prediction compared with the other models. The trend also happens in the respiratory model, in which the MC boxplot indicates the smallest range of interquartile. However, the models will keep changing as the new data arrive. Therefore, for production implementation, it needs dynamics retraining. The methods can vary, such as triggering the retraining by regular scheduling or a particular RMSE threshold calculation.

## 6. Conclusions and Future Works

The influenza-like illness and respiratory disease were predicted by comparative models of ARIMA and LSTM for 5 and 10 year periods. While ARIMA captured a variety of typical temporal structures of the diseases that change over time in the training model, LSTM carried out some feature selection techniques, including matrix correlation, extra trees classifier, and chemical-based before training input data. As a result, three LSTM-based training types were formed, two of which selected the top five highest-ranking data. The remaining type, chemical-based, used four parameters, which are the main contributors to the diseases. Thus, the study conducted comparisons of the disease’s prediction between ARIMA, ordinary LSTM, and three LSTM models associated with feature selection techniques during five years and ten years. As a result, there is a highly consistent trend of the prediction models for ILI and respiratory cases. Besides, all four LSTM prediction models present a higher accuracy than ARIMA’s outcome in both 5 and 10 year periods of ILI and respiratory diseases. Comparing two periods of time, the prediction model of ARIMA for five years is more accurate than one for ten years, whereas the ten-year prediction models using LSTM got the opposite result, the result of ten-year prediction models reach a higher accuracy than five-year models’. Among the four LSTM-based models, the matrix correlation technique’s outcomes achieve the most accuracy, and chemical-based feature selection is also a promising approach. The chemical-based show is not much different from the matrix correlation’s results, and its four selected factors are based on fundamental knowledge.

In the future, the model’s implementation can be applied in further development such as ARIMAX to provide the balance input parameters for a statistically-based model. Also, the models can be integrated along with the sensor. Therefore, a real-time response system can be activated to mitigate disease outbreaks, such as the recent COVID-19 pandemic infection in advanced time.

## Figures and Tables

**Figure 1 ijerph-19-01858-f001:**
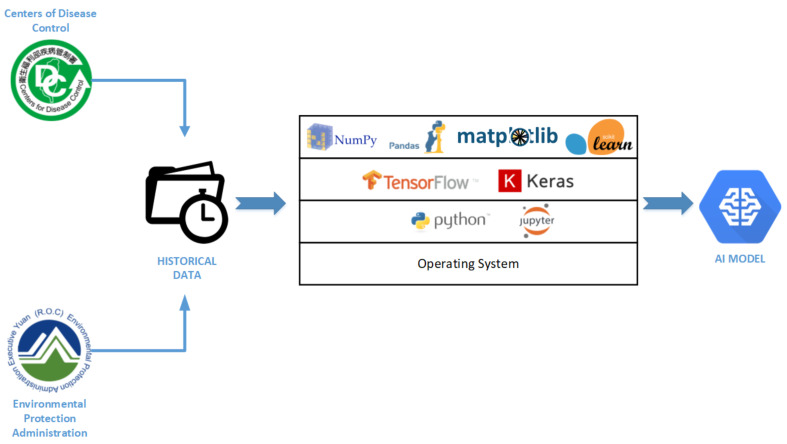
System Architecture.

**Figure 2 ijerph-19-01858-f002:**
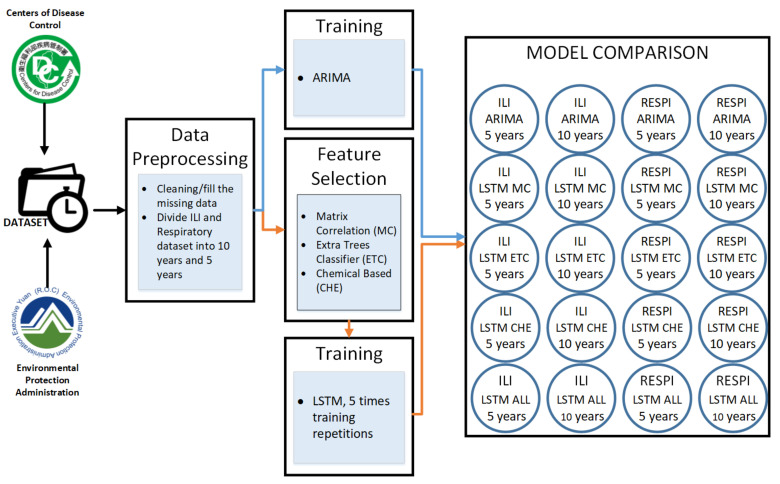
System Workflow.

**Figure 3 ijerph-19-01858-f003:**
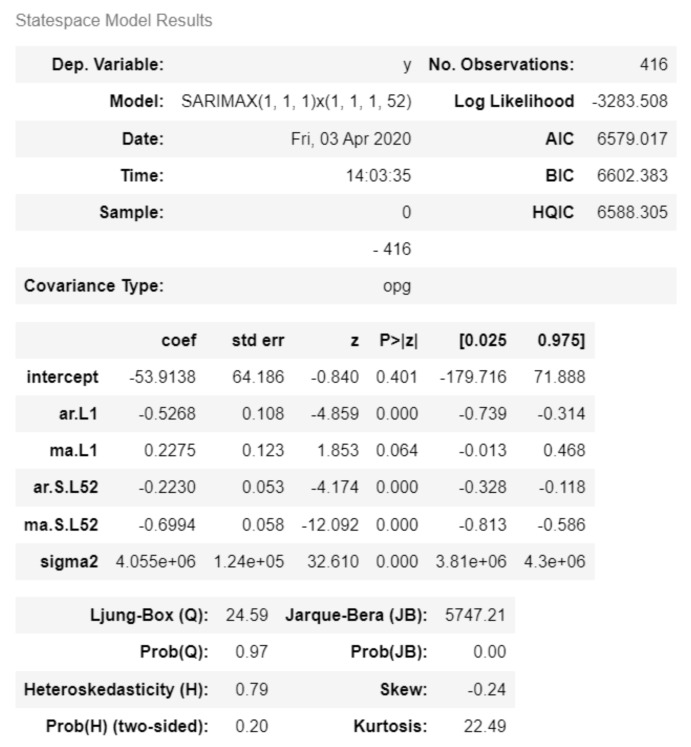
ARIMA model statespace.

**Figure 4 ijerph-19-01858-f004:**
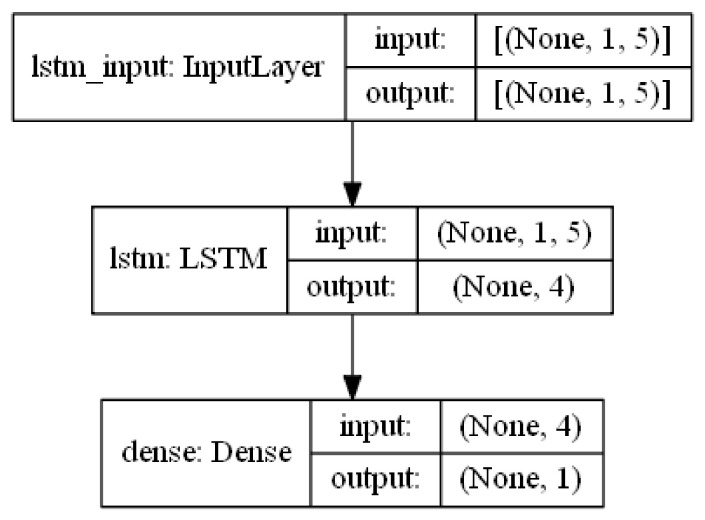
LSTM model architecture.

**Figure 5 ijerph-19-01858-f005:**
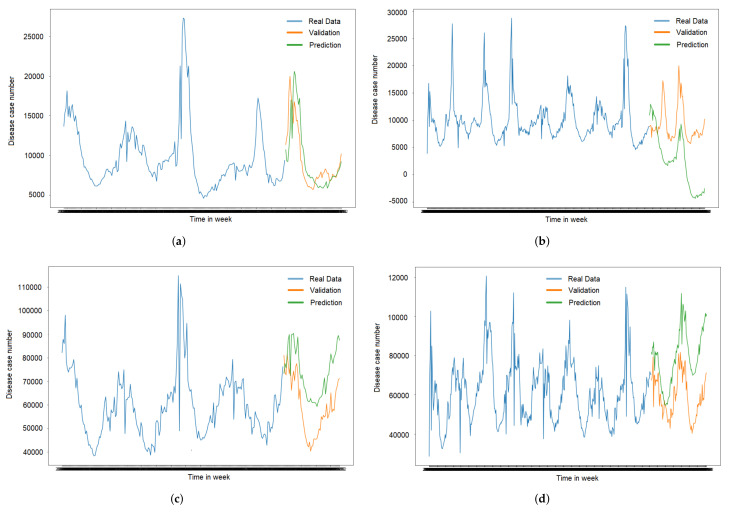
ILI and Respiratory ARIMA models. (**a**) ILI from 5 years dataset training; (**b**) ILI from 10 years dataset training; (**c**) Respiratory from 5 years dataset training; (**d**) Respiratory from 10 years dataset training.

**Figure 6 ijerph-19-01858-f006:**
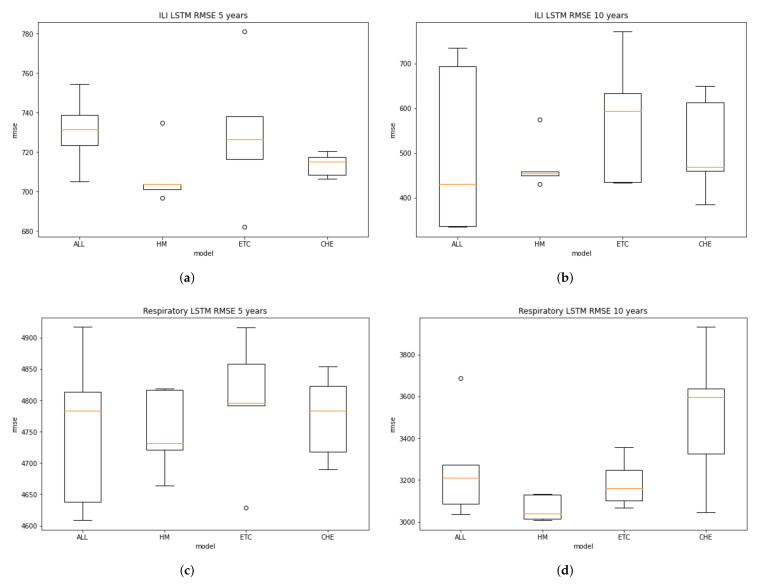
The bloxplot prediction distribution in 5 times repetitions. (**a**) ILI LSTM boxplot from 5 years dataset training; (**b**) ILI LSTM boxplot from 10 years dataset training; (**c**) Respiratory LSTM boxplot from 5 years dataset training; (**d**) Respiratory LSTM boxplot from 10 years dataset training.

**Figure 7 ijerph-19-01858-f007:**
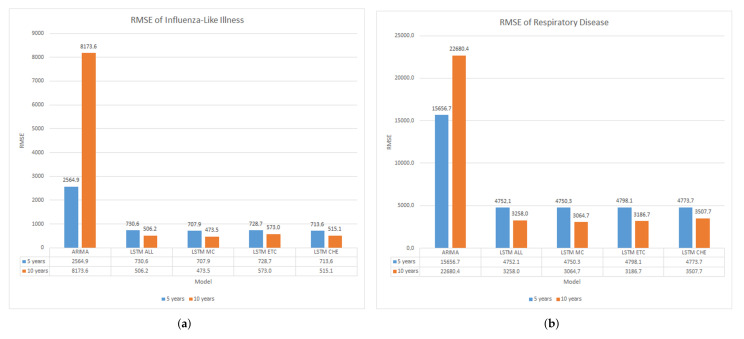
The model comparison results in graphs. (**a**) The RMSE of Influenza-like Illness; (**b**) The RMSE of Respiratory Disease.

**Figure 8 ijerph-19-01858-f008:**
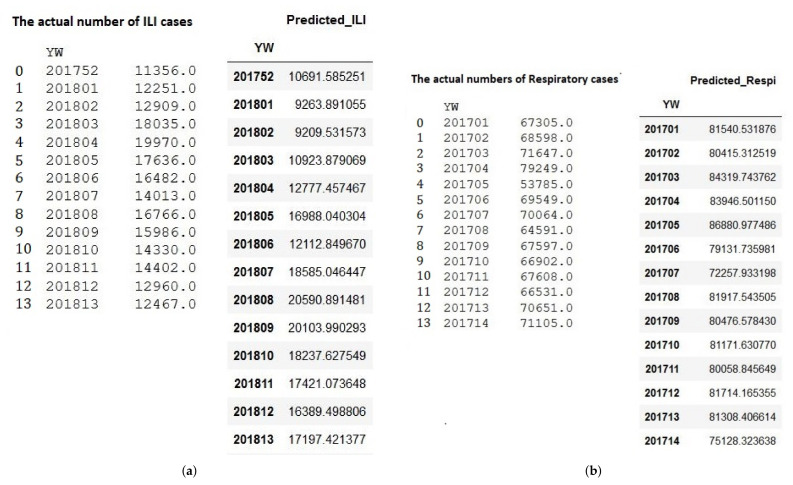
The Prediction sampling of ARIMA. (**a**) ARIMA ILI 5 years; (**b**) ARIMA Respiratory 10 years.

**Figure 9 ijerph-19-01858-f009:**
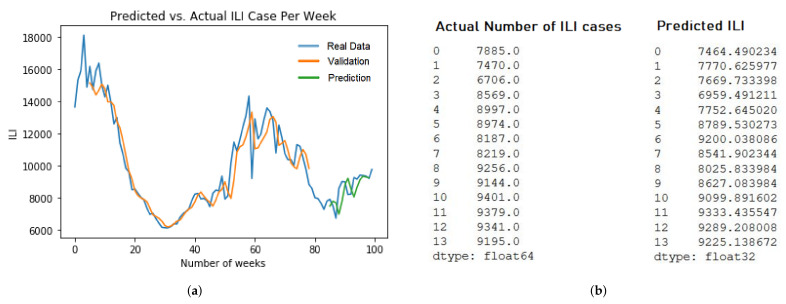
ILI LSTM sampling predictions. (**a**) ILI LSTM MC model for 10 years dataset training; (**b**) The Prediction Result of LSTM MC 10 years.

**Figure 10 ijerph-19-01858-f010:**
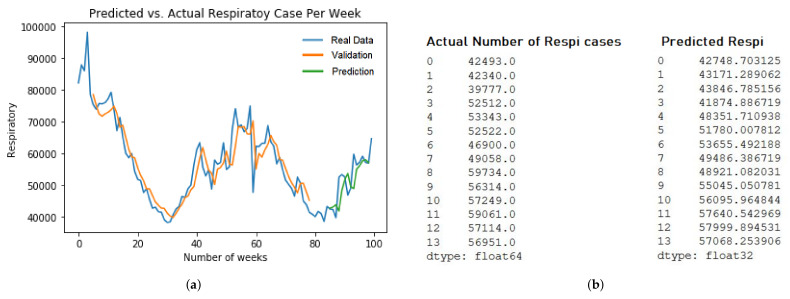
Respiratory LSTM sampling predictions. (**a**) Respiratory LSTM ALL model for 5 years dataset training; (**b**) The Prediction Result of LSTM ALL 5 years.

**Table 1 ijerph-19-01858-t001:** Matrix Correlation of ILI and Respiratory Disease.

Order	ILI	Respiratory
1	AMB_TEMP	AMB_TEMP
2	NOx	NO2
3	NO2	NOx
4	NO	CO
5	CO	NO

**Table 2 ijerph-19-01858-t002:** Extra Trees Classifier of ILI and Respiratory.

Order	ILI	Respiratory
1	NO2	NOx
2	O3	WIND_SPEED
3	WIND_DIREC	AMB_TEMP
4	SO2	PM10
5	NOx	WS_HR

**Table 3 ijerph-19-01858-t003:** Chemical Base of Feature Selection.

Order	Correlation
1	PM2.5
2	PM10
3	NO2
4	SO2

**Table 4 ijerph-19-01858-t004:** Dataset Air Pollutant Parameters.

**Chemical Pollutant Factors**			
**No.**	**Parameter**	**Unit**	**Name**
1	CO	ppm	Carbon Monoxide
2	NO	ppb	Nitrogen Monoxide
3	NO2	ppb	Nitrogen Dioxide
4	NOx	ppb	Nitrogen Oxide
5	O3	ppb	Ozone
6	PM2.5	μg/$m^{3}$	fine aerosol
7	PM10	μg/$m^{3}$	aerosol
8	SO2	ppb	Sulfur Dioxide
**Meteorogical Factors**			
1	AMB_TEMP	Celsius	ambient temperature
2	RAINFALL	mm	Rainfall
3	RH	%	Relative Humidity
4	WD_HR	degrees	Wind Direction Hourly
5	WIND_DIREC	degrees	Wind Direction
6	WIND_SPEED	m/s	Wind Speed
7	WS_HR	m/s	Wind Speed Hourly

**Table 5 ijerph-19-01858-t005:** The RMSE of ARIMA Model Comparison.

ARIMA Model	RMSE
ILI ARIMA 5 years	2174.8
ILI ARIMA 10 years	8173.6
Respiratory ARIMA 5 years	5581.6
Respiratory ARIMA 10 years	15,093.1

**Table 6 ijerph-19-01858-t006:** ILI RMSE using 5 and 10 years data.

**Training** **Repetitions**	**ILI LSTM RMSE 5 Years**			
	**ALL**	**MC**	**ETC**	**CHE**
1	754.3	703.6	780.9	706.4
2	705.1	701.0	682.1	717.4
3	738.8	703.6	738.0	715.2
4	723.3	696.7	726.3	708.5
5	731.5	734.8	716.3	720.4
**mean**	**730.6**	**707.9**	**728.7**	**713.6**
**std**	**18.3**	**15.3**	**35.9**	**5.9**
** Training** **Repetitions**	**ILI LSTM RMSE 10 Years**			
	**ALL**	**MC**	**ETC**	**CHE**
1	336.19	574.67	433.57	459.57
2	734.85	449.82	594.45	385.61
3	431.1	458.10	633.23	612.43
4	335.31	430.23	770.90	649.33
5	693.32	454.50	435.67	468.72
**mean**	**506.2**	**473.5**	**573.6**	**515.1**
**std**	**194.3**	**57.6**	**142.8**	**111.2**

**Table 7 ijerph-19-01858-t007:** Respiratory RMSE using 5 and 10 years data.

**Training** **Repetitions**	**RESPIRATORY LSTM RMSE 5 Years**			
	**ALL**	**MC**	**ETC**	**CHE**
1	4813.9	4731.2	4629.1	4690.1
2	4637.6	4817.0	4916.0	4783.7
3	4608.9	4720.7	4858.2	4853.7
4	4916.9	4818.8	4795.5	4718.4
5	4783.1	4663.6	4791.6	4822.4
**mean**	**4752.1**	**4750.3**	**4798.1**	**4773.7**
**std**	**128.0**	**66.9**	**107.4**	**68.7**
**Training** **Repetitions**	**RESPIRATORY LSTM RMSE 10 years**			
	**ALL**	**MC**	**ETC**	**CHE**
1	3273.5	3015.2	3355.9	3046.1
2	3084.6	3130.4	3160.6	3596.6
3	3210.2	3131.0	3247.6	3326.3
4	3687.0	3007.9	3066.8	3931.4
5	3034.6	3038.8	3102.6	3638.3
**mean**	**3258.0**	**3064.7**	**3186.7**	**3507.7**
**std**	**258.1**	**61.4**	**116.7**	**335.6**

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
