# Peer review of "The Prediction of Influenza-like Illness and Respiratory Disease Using LSTM and ARIMA"

_ijerph, 2022, doi:10.3390/ijerph19031858_

Round 1
Reviewer 1 Report
I think the paper titled “The Prediction of Influenza-Like Illness and Respiratory Disease Using LSTM and ARIMA” is a good research paper and deserves to be published in this journal. However, the following minor comments concerns should be considered before publication:
- All grammar errors should be corrected.
- For instance, in the abstract, it is written that "Second, we implemented the LSTM method, which trained based on the correlation between the weekly number of disease and ..." should be corrected as "Second, we implemented the LSTM method, which trained based on the correlation between the weekly number of diseases and ...". Again, in the abstract, it is written "... LSTM has better accuracy in ten years dataset compare to five years dataset " should be corrected as "LSTM has better accuracy in ten years dataset compared to five years dataset. " .
- Figures (3) - (5) are too small. Also, more comments may be added for these Figures.
- Is it possible to make some comments with the dataset of hundred years terms?
- In the Conclusions and Future Works, it is written that "Therefore, a real-time response system can be activated to mitigate the outbreaks disease, such as ... " should be corrected as " Therefore, a real-time response system can be activated to mitigate the outbreaks of disease, such as ... "
Author Response
Dear editors and reviewers,
Thank you for giving the constructive comments to enhance the manuscript entitled “The Prediction of Influenza-Like Illness and Respiratory Disease Using LSTM and ARIMA” (Paper# ijerph-1548855). It is our pleasure to have the second evaluation for publication. The authors have been considered all reviewers’ suggestions, and the manuscript provides higher readability and completeness in the reversion. The authors would like to provide the revised manuscript and the comment reply, and all modifications in the manuscript are marked by yellow. We hope that the current version is qualified to be considered for publication in the MDPI IJERPH.
Sincerely Yours,
Authors

Reviewer 2 Report
Forecasting a type of disease is very important in public health. This article presented a comparison of two forecasting methods for influenza-like illness and respiratory disease. This article appears to have some limitations, which include that:
- The article is lack of methodological originality. ARIMA and LSTM are well-known techniques and many studies already showed that LSTM tends to outperform ARIMA. Also, the article could provide methodological description for the methods.
- Subject to the first comment, the conclusion/contribution is not strong enough. The article could provide significant scientific contribution and implication of study with respect to forecasting influenza-like illness and respiratory disease.
- The study design seems not sound because the ARIMA model does not use other input variables while the LSTM uses them, which leads to apple and orange comparison. The authors could use an ARIMAX model to incorporate other input variables.
Author Response

(The authors gave the same response as above.)

Reviewer 3 Report
First of all, congratulations for the work carried out, it is a very interesting research and the methodology of the study is very well defined and developed. After reviewing the manuscript, it could be improved in the following points:
The introduction justifies the study well but should include the existence or not of previous studies at national or international level.
The research objectives are well defined, but the main objective is stated as "to predict the amount case of ILI and respiratory disease". The discussion is brief and in my opinion should better develop the main objective mentioned.
Finally, I think it would be interesting to include in the manuscript a separate section on "limitations of the study".
Best regards.
Author Response

(The authors gave the same response as above.)

Reviewer 4 Report
The paper claims to propose a deep learning model to predict Influenza-like Illness (ILI) and Respiratory Disease using various models comparison and feature selection methods. In this work, LSTM and ARIMA models are applied for predicting Influenza-Like Illness and Respiratory Disease. The obtained feature models are used for training through LSTM and ARIMA. The average RMSE of the work is evaluated and compared using a time-span of 5years dataset and 10 years dataset. The flowing points must be included:
- The introduction section needs modification. The objectives of the work should be explained in detail.
- Authors must separate related work from Introduction. The authors have not mentioned their real drawbacks or the need of the proposed model. This section must include more recent works.
- Section Material and Methods: Authors need to justify the use of LSTM and ARIMA deep learning models.
- The author has to include the details about layers used in LSTM (Section 2.5.2) and ARIMA (Section 2.5.1) with precise formulas and filters.
- The author has to include the details about the Implementation environment and tool details used for the proposed work.
- The author has to include the equations of classifier performance like RSME.
- The results discussion and justification of the findings need to be included and detailed for each table and figure. Tables 5-8 and figures 7-18.
- You have to define the acronyms used in the article, such as LSTM and ARIMA
In conclusion, authors must include recent related works, more details about the proposed method, implementation details, and the results discussion and justification of the findings need to be included and detailed
Author Response

(The authors gave the same response as above.)

Round 2
Reviewer 2 Report
The revised version update some writing parts including literature review, but did not change most of the analytical results.
- Feature selection is made before applying LSTM, which may not guarantee the best feature selection. Additionally, the sparsity hyperparameter may control the effects of input variables further.
- It is not clear if the authors used the same training (80%) and test (20%) data sets for both ARIMA and LSTM. The dates corresponding to those 80% and 20% should be specified.
- The authors did not provide a clear step for handling autoregressive terms of ILI or Respiratory data. It was not stated how many AR terms were used. If AR terms were not modeled, the LSTM model may not be a time series model.
- Visualization of results including tables and figures could be improved. Some tables could be combined. Copy and paste of results from a program code does not seem professional. Some side-by-side tables have different table sizes and font sizes. Tables 6-7 and Figure 6 have the same information, which may be redundant. No legends are available for Figures 8 and 9.
- English writing should be properly addressed.
Author Response
Dear editors and reviewers,
Thank you for giving the constructive comments to enhance the manuscript entitled “The Prediction of Influenza-Like Illness and Respiratory Disease Using LSTM and ARIMA” (Paper# ijerph-1548855). It is our pleasure to have further evaluation for publication. The authors have been considered all reviewers’ suggestions, and the manuscript provides higher readability and completeness in the reversion. The authors would like to provide the revised manuscript and the comment reply, and all modifications in the manuscript are marked by yellow. We hope that the current version is qualified to be considered for publication in the MDPI IJERPH.
Sincerely Yours,
Chao-Tung Yang Ph.D.

Reviewer 4 Report
The authors answered all my questions in a clear and understandable way.
I recommend accepting this paper.
Author Response
Dear reviewer,
Thank you for giving the constructive comments to enhance the manuscript entitled “The Prediction of Influenza-Like Illness and Respiratory Disease Using LSTM and ARIMA” (Paper# ijerph-1548855). It is our pleasure to have further evaluation for publication.
Sincerely Yours,
Chao-Tung Yang, Ph.D.